# The Mediating Role of Classroom Climate on School Violence

**DOI:** 10.3390/ijerph18062790

**Published:** 2021-03-10

**Authors:** David Montero-Montero, Paula López-Martínez, Belén Martínez-Ferrer, David Moreno-Ruiz

**Affiliations:** 1Department of Education and Social Psychology, Pablo Olavide University, 41013 Seville, Spain; pmlopmar@upo.es (P.L.-M.); bmarfer2@upo.es (B.M.-F.); 2Department of Social Psychology, Valencia University, 46010 Valencia, Spain; david.moreno-ruiz@uv.es

**Keywords:** classroom climate, peer victimization, peer aggression, delinquency, mediation analysis

## Abstract

Mediation analysis has been confirmed as a very useful statistical tool in the social sciences, especially in school-related studies. This type of analysis was used in the present study to examine the mediating role of classroom climate (measured with the classroom environmental scale), categorized into three dimensions, namely involvement, affiliation, and teacher support, on the relationship between peer victimization and peer aggression. The participants consisted of 2011 adolescents (50.67% boys and 49.32% girls), aged between 12 and 18 years old (M = 14.17; SD = 1.47), enrolled in schools in Andalusia (Spain). Findings revealed a significant direct relationship of all the dimensions. They also revealed that teacher support was the only dimension that mediates in the relationship between peer violence and peer aggression. The results and their implications for improving classroom climate and school violence are discussed.

## 1. Introduction

Since the first research in the field of peer aggression in the 1970s [1], many researchers have confirmed the relevance of this problem, which has great implications for the psychosocial adjustment of adolescents [2,3]. More specifically, peer aggression is considered to be a public health issue [4] that affects the well-being of children and adolescents [5], and even the development of their adult life, making it a complex problem. Previous research has identified various risk factors [6,7], and personal and social adjustment problems [8,9], associated with peer aggression and peer victimization.

Since this type of violence takes place at school, the most relevant formal institution in the psychosocial development of minors [10], research into school factors is especially important. In fact, interactions between students and also between students and teachers develop in the school setting. These relationships generate a climate that can promote the development and maintenance of maladjusted behaviours in minors. This study examines the mediating effect of classroom climate on the relationship between peer victimization and peer aggression in determining whether it can be related to the fact that young people who are peer victimized may develop behaviours associated with peer aggressors. 

### Peer Victimization, Peer Aggression, and Classroom Climate

Classroom climate is deemed to refer to the subjective meanings that students have in the classroom where they study, encompassing dimensions such as the degree of involvement and motivation of students (involvement), cohesion between students (affiliation), and the relationship between the teacher and the student (teacher support) [11,12,13]. The interest of this factor lies in the subjective element of the information obtained on students’ own perceptions of interpersonal relationships, commitment, and motivation at school [12,14]. Furthermore, both low academic engagement [15] and low perceived teacher support by students have been related to disruptive and violent behaviours at school [16]. This dimension of classroom climate, namely teacher support, is related to formal authority, whose relevance has also been widely confirmed in the study of peer aggression [17].

A positive classroom climate has been confirmed as an indicator of education quality [18]. In addition, the dimensions that make up classroom climate, such as personal involvement and affiliation between peers, can favour or harm the normal psychosocial development of children and adolescents [19]. Previous research supports the hypothesis that these dimensions may act as risk factors due to their direct link with low levels of self-esteem and empathy [10,20].

The joint perception of these components allows a distinction to be made between positive and negative climates [12]. Obtaining good academic grades, perceiving classmates as friends, teacher support, and peer affiliation, as well as having a good evaluation of the school, are aspects of a positive classroom climate. This positive assessment helps students achieve a better psychosocial adjustment and thus a lower risk of participating in episodes of peer aggression [21,22]. In contrast, a low level of connection and a largely ungratifying perception of school are related to a negative classroom climate, which favours participation in peer aggression, and also increases the risk of being victimized at school [12,21], thus facilitating the simultaneous appearance of the figure of both bully and victim.

The importance of classroom climate in the violence–victimization dynamic is a key factor that requires further study. Therefore, the main objective of this study was to examine the mediating role of the different dimensions of classroom climate (involvement, affiliation, and teacher support) in the relationship between peer victimization and peer aggression. With this objective in mind, four hypotheses were developed based on the model illustrated in Figure 1.

**Hypothesis** **1** **(H1).**
*Peer victimization would be positively correlated with peer aggression.*


**Hypothesis** **2** **(H2).**
*Peer victimization would be negatively correlated with classroom climate.*


**Hypothesis** **2a (H2a).**
*Peer victimization would be negatively correlated with the involvement dimension of classroom climate.*


**Hypothesis** **2b (H2b).**
*Peer victimization would be negatively correlated with the affiliation dimension of classroom climate.*


**Hypothesis** **2c (H2c).**
*Peer victimization would be negatively correlated with the teacher support dimension of classroom climate.*


**Hypothesis** **3** **(H3).**
*Classroom climate would be negatively correlated with peer aggression.*


**Hypothesis** **3a (H3a).**
*Peer victimization would be negatively correlated with the involvement dimension of classroom climate.*


**Hypothesis** **3b (H3b).**
*Peer victimization would be negatively correlated with the affiliation dimension of classroom climate.*


**Hypothesis** **3c (H3c).**
*Peer victimization would be negatively correlated with the teacher support dimension of classroom climate.*


**Hypothesis** **4** **(H4).**
*The dimensions of classroom climate would have a mediating effect on the relationship between peer aggression and peer victimization.*


## 2. Materials and Methods 

### 2.1. Participants and Procedure

The data obtained in this study were obtained from a sample of 2011 adolescent students (50.67% boys and 49.32% girls) selected from 4 secondary schools (2 public and 2 state-subsidized) in Andalusia (Spain). The participants were aged between 12 and 18 (M = 14.47, SD = 1.47). Participants were selected randomly. A sampling error of ±2.5% and a confidence level of 95% were established, with an expected population variance of 0.50. 

Initially, the selected educational centres were contacted to request their participation and explain the objectives of the present study. Subsequently, the participating centres sent letters to the students’ parents, explaining the research and requesting the corresponding written consent to allow their children to participate; only 0.99% of the sample (n = 20) rejected the invitation to participate. Then, the instruments were administered in a session of approximately 45 minutes. The instruments were applied in the regular classes of the participating groups and during an ordinary class period. The participants were informed that their participation in the study was voluntary and anonymous, and that they could stop participating at any point in the process.

Additionally, the study complied with the ethical values required in research with human beings and respects the fundamental principles included in the Helsinki Declaration [23] and subsequent updates thereto. 

### 2.2. Instruments

The peer aggression scale was used to assess violent behaviour at school [24,25]. This scale comprises 25 items on a 4–point Likert-type scale (1 = never and 4 = always), where students analyze and report their aggressive behaviour towards other students (e.g., When someone makes me angry, I harm or hurt them). Confirmatory factor analysis (CFA) showed a good fit of the model to the measured data (SB Х^2^ = 725.2593, df = 233, *p* < 0.001, CFI = 0.915, RMSEA = 0.030, I.C. 90 (0.027, 0.032)). The reliability coefficient of Cronbach’s alpha was 0.90. The validation of the scale has been confirmed in previous literature [24,26].

In the case of victimization, the peer victimization scale [24,27] was used. This scale comprises 20 items in which the students analyze and report having been victims of violent situations (e.g., A classmate has made fun of me to really upset me). Confirmatory factor analysis (CFA) showed a good fit of the model to the measured data (SB Х^2^ = 664.279, df = 159, *p* < 0.001, CFI = 0.932, RMSEA = 0.036 (0.034, 0.039)). The Cronbach alpha reliability coefficient was 0.87. 

Classroom climate was measured using the classroom environment scale (CES) [20,28]. This scale comprises 30 items, with two answer options (True/False): on social climate and interpersonal relationships between the components of the student group. This instrument measures three different scales: (1) involvement (e.g. “Students show a lot of interest in what they do in class”); (2) affiliation (e.g. “In class, students get to know each other well”); and (3) teacher support (e.g., “The teacher shows interest in his/her students”). Cronbach’s alpha indicates that the reliability of these sub-scales is 0.84, 0.79, and 0.89, respectively, and that of the general scale 0.85. 

### 2.3. Statistical Analysis

The main aim was to examine the mediating role of the different dimensions of classroom climate (involvement, affiliation, and teacher support) on the relationship between peer victimization and peer aggression. First, descriptive statistics (rank, minimum, maximum, mean, and standard deviation) were obtained. All measures were standardized. Then, to achieve the main goal of the study, a simple mediational model (Model 4) with 10,000 bootstraps using the macro PROCESS [29] in SPSS (Version 25.0, IBM, Armonk, NY, USA) was performed. An 95% confidence interval (CI) was provided as an estimate. As for the mediational size effect, if the 95% CI of the index does not include zero, the index of the mediation is significant [30].

## 3. Results

### Mediation Analyses

Descriptive results can be found in Table 1. In terms of direct significant relationships, first, peer victimization as a predictor of the classroom climate dimensions was examined. As shown in Figure 2, peer victimization was negatively related to school involvement (ß = −0.20, SE = 0.02, t = − 9.03, *p* < 0.001, 95% CI = (−0.24, −0.15)), affiliation (ß = −0.25, SE = 0.02, t = −11.54, *p* < 0.001, 95% CI = (−0.29, −0.21)) and teacher support (ß = −0.14, SE = 0.02, t = −6.54, *p* < 0.001, 95% CI = (−0.18, −0.10)). On the other hand, the three dimensions of classroom climate were directly and negatively related to peer aggression: school involvement (ß = −0.05, SE = 0.02, t = −2.10, *p* < 0.05, 95% CI = (−0.09, 0.00)), affiliation (b = −0.05, SE = 0.02, t = 2.21, *p* < 0.05, 95% CI = (−0.10, −0.01)) and teacher support (b = −0.12, SE = 0.02, t = −5.30, *p* < 0.001, 95% CI = (−0.17, −0.08)) (See Figure 2). Last, peer victimization was positively and significantly related to peer aggression (b = 0.19, SE = 0.02, t = 863, *p* <.001, 95% CI = (0.15, 0.24)) (See Figure 2). 

As regards the indirect effect, when school involvement, affiliation, and teacher support were considered, peer victimization was still positively and significantly related to peer aggression via teacher support (ß = 0.23, SE = 0.02, t = 10.73, *p* < 0.001, 95% CI = (0.19, 0.27)). Therefore, peer victimization was related to a lower perception of teacher support, which was also related to higher levels of school violence (See Figure 2). 

To assess the effect size of the mediation paths, the standardized indirect effect was used [31,32]. Results did not yield a significant indirect effect of involvement (effect = 0.01, SE 0.00, 95% CI (0.00–0.02)) or affiliation (effect = 0.01, SE = 0.01, 95% CI (0.00–0.03)). However, as expected, a significant indirect effect of teacher support was observed on the relationship between peer victimization and peer aggression (effect = 0.02, SE = 0.00, 95% IC (0.01–0.03)) and the percentage of explained variance was 9.88%. Therefore, a direct effect of peer victimization on peer aggression, and a mediational effect via teacher support, were observed, indicating that peer victimization is associated with a lower perception of teacher support, which is also negatively related to peer aggression. 

## 4. Discussion

Peer violence at school is an important problem that affects thousands of young people around the world [31]. Nevertheless, much remains to be learned about the issues that mediate the relationship between peer violence and peer aggression.

Therefore, the main aim of this study was to examine the mediating role of the different dimensions of classroom climate (involvement, affiliation, and teacher support) in the relationship between peer victimization and peer aggression. To arrive at this point, we confirmed that there was a positive relationship between peer victimization and peer aggression (H1).

In line with previous evidence, a negative relationship between peer victimization and classroom climate (involvement, affiliation, and teacher support) was identified (H2). This would imply that higher peer victimization leads to lower involvement (H2.a), lower peer affiliation (H2.b), and lower perceived teacher support (H2.c).

Similarly, a negative relationship was found between the three dimensions of classroom climate (involvement, affiliation, and teacher support) and aggression, as hypothesized (H3). These results indicate that lower involvement (H3.a), poorer peer relations (H3.b), and lower perceptions of teacher support (H3.c) may be related to an increase in peer aggression. Thus, students who report higher peer victimization also report worse perceptions of classroom climate and higher peer aggression, which is consistent with previous research [12,20].

In terms of the mediation analysis, this study supports the idea that the perception of teacher support exerts a mediating effect between peer victimization and peer aggression. Despite the direct link identified, two of the three dimensions of classroom climate, involvement and affiliation, did not have a significant indirect effect between peer victimization and peer aggression, thus refuting H4 almost entirely. However, the teacher support dimension was significant. This relationship suggests that higher peer victimization is related to lower teacher support, which is, in turn, associated with higher peer aggression. This relationship could partially explain why some young people involved in school aggression develop more serious psychosocial adjustment problems, due to all the time they spend in a space where they feel unprotected, do not perceive affiliation or teacher support, and do not perceive sufficient participation [32,33]. In this context of poor perception of protection, some studies indicate that victimized students may find violence and revenge as a solution for their situation [34], so it may be interesting to incorporate variables in this regard in future research.

The interpretation of these relationships between the dimensions of classroom climate and the processes of violence-victimization has relevant implications for understanding the psychosocial dynamics that occur in the classroom. More specifically, the role of the teacher must be highlighted. Teachers with the necessary training could help reduce episodes of peer aggression by promoting a better perception of the climate through support for students. This support can be achieved by increasing the students’ perception of whether their opinion is heard and their participation valued [35], and fostering a more collaborative and less competitive school system, which could help to strengthen ties between students (affiliation). 

Direct intervention in the classroom climate not only fosters a better environment in the students’ day-to-day life and a reduction in peer aggression episodes. It is also indirectly related to other factors and contexts. For example, a greater perception of affiliation with peers is linked to greater empathy, which encourages the expression of feelings in the family system [36]. In addition to the teacher support dimension of the classroom climate, other variables not included in this study may also exert a mediating effect between peer victimization and peer aggression; hence, the need to continue research in the field of peer aggression.

### Limitations

The present study provides relevant information for improving classroom climate, since specific elements are identified that can reinforce its positive perception. In this case, teacher support. However, it is important to bear in mind that this is a cross-sectional study, which, by definition, does not allow the establishment of causal relationships or the identification of behavioural patterns. Bidirectional influences should also not be ruled out, given the nature of the variables. The aforementioned complexity of school violence is evidenced precisely through its multi-factorial relationship [6,26,37]. To establish causal relationships between the dimensions analyzed in this study, it would be necessary to carry out a longitudinal study in different periods of time. 

In terms of the measurement tool used for the classroom climate variable, it is important to remember that the use of self-reports, despite constituting an important source of data in this type of study, may have certain limitations due to biases such as, for example, social desirability. However, it would be of interest to contrast the information obtained in the self-report surveys with the information provided by other sources, such as parents and teachers. 

In any case, the mediating effect of other variables that have not been studied in this research must be considered, as well as verifying whether the results are replicated at other educational levels, which may be taken into account for future research.

## 5. Conclusions

Classroom climate is an essential element when assessing educational quality. Given the importance of schools as an institution, knowledge in this respect is key to understanding the psychosocial development of young people and adolescents [38].

This study examined the mediating role exerted by classroom climate on the relationship between peer victimization and peer aggression, and a direct effect was observed between all the dimensions comprising classroom climate, namely involvement, affiliation, and teacher support; but indirect effect was only identified on teacher support. The importance of the results obtained is twofold: on the one hand, by focusing on subjective elements through a self-report survey, the relationship is certainly relevant; on the other, since information from schools is analyzed, it can be used as a basis for applying intervention elements to improve educational coexistence.

The implications of these findings are important and, if applied correctly, can help promote a better classroom climate. In turn, a better perception of classroom climate may favour a reduction in peer victimization and peer aggression, which would have fewer negative implications in the psychosocial development of younger people in the population. Taking into account the results obtained, the figure of the teacher should be reinforced to allow them to dedicate more time to students, and thus favour greater cohesion among students and greater personal involvement in day-to-day obligations and responsibilities in school. 

## Figures and Tables

**Figure 1 ijerph-18-02790-f001:**
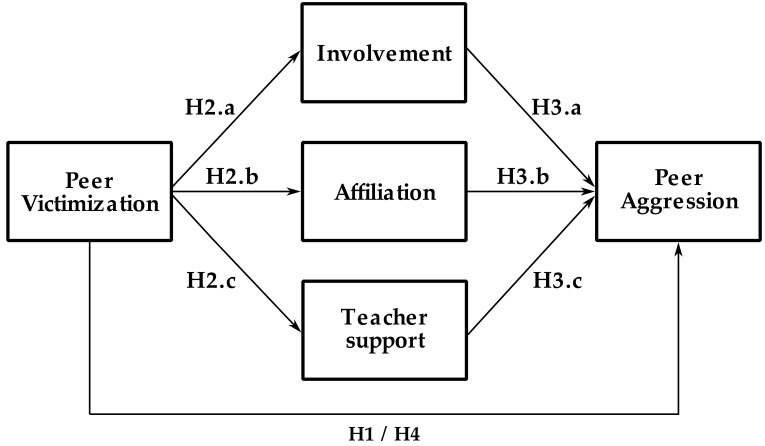
Illustration of the proposed mediation model**.**

**Figure 2 ijerph-18-02790-f002:**
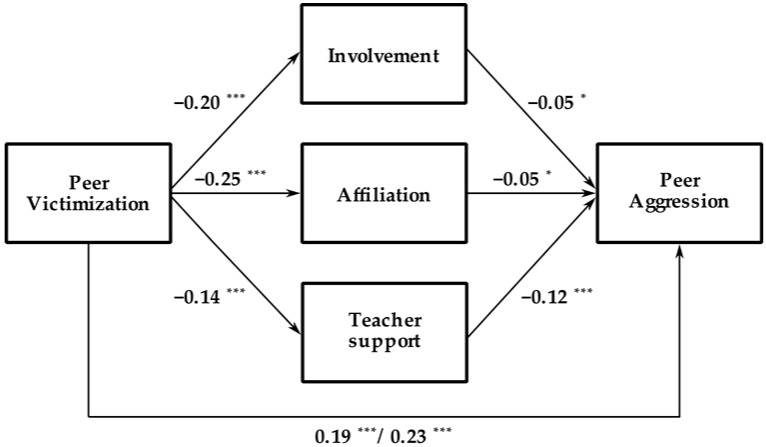
Illustration of the resulting model with the values obtained in the analysis of the mediation role of the three classroom climate dimensions on the relationship between peer victimization and peer aggression. *: *p* < 0.05, ***: *p* < 0.001.

**Table 1 ijerph-18-02790-t001:** Rank, minimum, maximum, and average values, and standard deviation.

Variables	Rank	Min.	Max.	M	SD
1. Peer Aggression	1.78	1.00	2.78	1.396	0.256
2. Peer Violence	2.66	1.00	3.66	1.412	0.345
3. School Involvement	1.00	1.00	2.00	1.447	0.204
4. Affiliation	0.90	1.10	2.00	1.720	0.169
5. Teacher Support	1.00	1.00	2.00	1.592	0.220

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
