# Peer review of "The Mediating Role of Classroom Climate on School Violence"

_ijerph, 2021, doi:10.3390/ijerph18062790_

Round 1

Reviewer 1 Report

This study aims to explore the importance of classroom climate in the violence-victimization dynamic by examining the mediating role of the different dimensions of school climate in the relationship between peer victimization and peer aggression. The key message is very important, but significant revisions seem to be needed.

Introduction

  1. Peer victimization, peer aggression and classroom climate were all measured cross-sectionally, so it should be impossible to determine causality as the authors describe in the Limitation. However, the analysis model is built on the assumption that peer victimization precedes the perception of classroom climate and peer aggression. In the Introduction, the theoretical backgrounds on which such an assumption is based should be clarified. These backgrounds do not seem to be sufficiently described.
  2. p.2, L47-61. These paragraphs refer to school climate. Although classroom climate and school climate seem to be treated almost identically, a distinction needs to be made between the two.
  3. p.2, L70-76. It has been hypothesized that the three dimensions of classroom climate are similarly related to the peer victimization and peer aggression, but the focus is mainly on the teacher support in the Introduction. If the authors assume that the three dimensions are equally relevant, then they need to focus equally on each. Alternatively, if the focus is specifically on teacher support, then the hypothesis needs to be modified as such.
  4. If the hypothesis 2 is for mediation analysis, it should be clarified whether direct or indirect effects, or both, from peer victimization to peer aggression are assumed.

Materials and Methods

  1. There is no description of Statistical Analysis. Please clarify the methods of mediation analysis.
  2. p.3, L110. Again, the differentiation between school climate and classroom climate is not clear.

Results

  1. The description of the results is incomprehensible. SE and t-value had better be omitted.
  2. The results for the direct and indirect effects of mediation analysis should be presented in a table.

Discussion

  1. p.4, L153-155. The authors stated “Hypothesis 1 was confirmed entirely”. Although significant correlations among peer victimization, peer aggression and classroom climate have been identified, the values of the standardized coefficients do not seem to be large, especially for between peer aggression and involvement, and affiliation. Also, an effect size of correlation between peer victimization and peer aggression association is large? Even if there were significant correlations, it does not mean that most of the victims have a tendency to perpetrate, so this point should be carefully stated to avoid misunderstandings.
  2. p.5, L157-159. It is not possible to draw this conclusion, as only the individual correlations between peer victimization and classroom climate, classroom climate and peer aggression, and peer victimization and peer aggression were found.
  3. p.5, L168-171. I didn't quite understand the meaning of this statement. Why does the fact that teacher support mediates the relationship between victimization and aggression partially explain the severity of psychosocial adjustment problems?

Conclusions

  1. p.6, L210-212. From the description of the Results, it is unclear whether there was a significant direct effect or not. In addition, the authors described “only an indirect effect was identified on teacher support”, but isn’t the key message of this study that teacher support is important because the students’ perception of teacher support mediated the relationship between peer victimization and peer aggression?

Reviewer 2 Report

Dear authors:

Abstract: The research findings need to be expanded in the summary 

Hypotheses: The hypotheses are very extensive, I would advocate making the hypotheses more precise also reflecting them in the "discussion" section 

Change of  “2.2. Materials” by “Instruments” In general, the article is well written and has cohesion and logic throughout it.

My congratulations to the authors for the novelty and adequacy of the article submitted.

Reviewer 3 Report

Thank you for the opportunity to review the manuscript entitled "The Mediating Role of Classroom Climate on School Violence".

The manuscript submitted for review examines a topic of great relevance in the field of educational psychology as is to know the mediating role of school climate on the relationship between peer victimization and peer aggression.

The topic is very important and the conceptual analysis made in the text is quite deep. The literature consulted is quite current and the sample is quite large (which is a strength for your work.). I would like to thank the efforts by the authors of the manuscript and congratulate them on the work. Overall, the writing is clear, the goals are well described, the introduction should explain the objectives of the study based on the review of the previous literature and the conclusions are properly made and presented. I consider that the constructs proposed in the abstract of the work are quite well explained. Therefore, the manuscript brings significant knowledge of the scientific literature so and still covers existing gaps in the field. On a formal level, the manuscript is perfectly structured, the references comply with the Journal rules. The work is ambitious and the results confirm the most of the hypotheses and the relevance and potential of the work is therefore recognized, but this Reviewer considers that the mediating role of other variables in school violence in addition to class climate should be taken into account as a work limitation, and they should indicate whether these results could be obtained at other educational levels. 

Round 2

Reviewer 1 Report

The authors have fully answered my proposal. Finally, it is better to check the English in the corrected parts.
